# Parallel adaptation and admixture drive the evolution of virulence in the grapevine downy mildew pathogen

Etienne Dvorak[1], Thomas Dumartinet[2], Isabelle D. Mazet[1], Alexandre Chataigner[1], Manon Paineau[3], Dario Cantù[3], Pere Mestre[4], Marie Foulongne-Oriol[5*‡], François Delmotte[1*‡]

1 INRAE, Bordeaux Sciences Agro, SAVE, ISVV, Villenave d'Ornon, France, 2 Université de Bordeaux, INRAE, BIOGECO, Cestas, France, 3 University of California Davis, Department of Viticulture and Enology, Davis, California, United States of America, 4 INRAE, Université de Strasbourg, SVQV, F, Colmar, France, 5 INRAE, MycSA, Villenave d'Ornon, France

‡ These authors are share senior authorship on this work.
* marie.foulongne-oriol@inrae.fr (MF-O); francois.delmotte@inrae.fr (FD)

## Abstract

Plant pathogens can rapidly adapt to host defenses, threatening the durability of resistance in crop varieties. It is thus crucial to identify the genetic determinants of virulence and understand how it arises and spreads in pathogen populations. In *Plasmopara viticola*, the biotrophic oomycete causing grapevine downy mildew, virulent strains have recently emerged following the deployment of cultivars carrying partial resistance factors. To investigate the genetic bases of adaptation to grapevine resistances, we carried out a QTL mapping study using two *P. viticola* biparental populations segregating for the ability to overcome two major loci, *Rpv10* and *Rpv12*. We identified the *AvrRpv12* locus, in which strains virulent towards *Rpv12* exhibited large homozygous deletions encompassing several RXLR effector genes. Population structure analyses further revealed that distinct alleles were selected independently in different winegrowing regions in Europe, highlighting multiple parallel adaptation events in response to resistance deployment. By contrast, the breakdown of *Rpv10* was determined by a dominant locus, suggesting an active suppressor mechanism. The virulent haplotype showed extensive structural rearrangements and a divergent effector repertoire. The locus corresponds to an admixed genomic segment likely originating from a recent secondary introduction of *P. viticola* into Europe. Beyond merely identifying candidate effectors, these results illustrate the range of evolutionary pathways through which pathogen populations adapt to plant resistances.

**Data availability statement:** New DNA sequences of *P. viticola* strains were submitted to NCBI SRA (BioProject PRJNA1255592). The genome assembly of strain Pv1419 as well as linkage mapping and phenotyping data produced in this study are available at https://doi.org/10.57745/KVUHYA.

**Funding:** This study was funded by grants to FD from the French National Research Agency (VITAE research program, agreement 20-PCPA-0010 and Plant2Pro® Carnot Institute, agreement 23-CARN-0024-01) and from the Horizon Europe scientific research initiative (GrapeBreed4IPM research program, agreement 101132223). TD was supported by the Bordeaux University program "Bordeaux Plant Sciences". The funders had no role in study design, data collection and analysis, decision to publish, or preparation of the manuscript. Funders websites https://anr.fr/,https://research-and-innovation.ec.europa.eu/,https://www.u-bordeaux.fr/.

**Competing interests:** The authors have declared that no competing interests exist.

## Author summary

Resistant varieties are a powerful method to control plant diseases, but they often lose efficiency upon the adaptation of plant pathogens. We aimed to understand the genetic bases of adaptation to plant resistances in *Plasmopara viticola*, an obligate parasite responsible for grapevine downy mildew, one of the most damaging diseases in viticulture. To do so, we crossed virulent *P. viticola* strains with avirulent ones and study how this trait was inherited in their progeny. We identified the genomics regions associated with virulence towards the resistance genes *Rpv10* and *Rpv12*. Both regions show high structural diversity and are enriched in genes encoding secreted proteins potentially interacting with the plant immune system. One of the virulences is recessive and the other dominant, suggesting different molecular mechanisms. In addition, we analyzed the genetic proximity between strains from different countries and found that the breakdown of *Rpv12* occurred multiple times independently in different winegrowing regions. By contrast, virulence towards *Rpv10* was probably the result of a new introduction of the pathogen in Europe. These results show how virulence can arise in plant pathogens populations through multiple mechanisms and evolutionary pathways.

## Introduction

Understanding how plant pathogens evolve to overcome host defenses is critical for the effective and sustainable management of crop diseases. Fungal and oomycete pathogens secrete a large number of effectors that promote infection [1,2], but some of them act as avirulence (Avr) factors when recognized by resistance (R) proteins [3]. Effector genes are thus evolving rapidly and evasion of the plant immune response can be achieved via multiple mechanisms, including non-synonymous mutations, deletions and silencing of Avr genes [4,5]. At the population level, the speed of adaptation depends on the ability of pathogens to generate genetic diversity and spread beneficial variants [6]. In particular, sexual reproduction plays a key role by creating new allele combinations and facilitating gene flow, leading to admixture between populations or even hybridization between species [7–9]. Therefore, in addition to the study of genetic determinants of plant-pathogen interactions, maximizing the durability of resistances requires the integration of population genetics approaches to elucidate the factors driving the emergence of virulence.

*Plasmopara viticola* is a biotrophic oomycete causing grapevine downy mildew, one of the most destructive diseases affecting vineyards worldwide. It was introduced from North America to Europe in the 1870s and from there spread to other winegrowing regions around the globe [10]. At the plot level, *P. viticola* populations are large and exhibit a high genotypic diversity [11]. Sexual reproduction occurs each year and outcrossing is ensured by strict heterothallism, resulting in a high heterozygosity rate

[12,13]. These elements contribute to a high evolutionary potential [6], as exemplified by the rapid loss of sensitivity to many fungicides [14,15].

Due to the high susceptibility of the cultivated Eurasian grapevine *Vitis vinifera*, breeding programs aiming to obtain resistant varieties are based on the introgression of resistances from wild grapes. Most of these genes only provide a partial protection, and their efficiency is variable depending on the environment, the plant physiological state and the genetic background of the variety [16]. Grapevines carrying major R factors trigger a hypersensitive response (HR) upon downy mildew infection, significantly limiting the pathogen's growth [17–19]. The best-studied loci encode proteins with nucleotide binding and leucine-rich repeat (NLR) domains, which are known for activating effector-triggered immunity (ETI) [20–22]. In particular, Rpv1 and Rpv3.1-mediated resistances were proved to be controlled by NLR genes [23,24], suggesting they act by recognizing Avr factors produced by *P. viticola*. Most oomycete Avr proteins possess an N-terminal RXLR motif, often associated with a dEER motif. Several RXLR(-like) effectors also share a modular structure mediated by a conserved fold called the (L)WY domain [25,26]. This type of putative effectors is abundant in *P. viticola*, as its genome contains more than 500 RXLR-like genes [12]. Oomycete effectors are often encoded in clusters of paralogous genes [27,28]. Such genes tend to be located in repeat-rich regions with high rates of duplication and deletion. This genomic compartmentalization is thought to foster an elevated diversity in the effector repertoire, which facilitates the rapid adaptation to plant defenses [29,30]. Several *P. viticola* secreted proteins have been found to interfere with the plant immune system [31–33], but specific interactions with major grapevine R genes were not investigated until recently.

Over the last few years, studies reported the widespread occurrence of *P. viticola* strains overcoming *Rpv3.1*, and the recent breakdown of *Rpv10* and *Rpv12*, two loci introgressed from the East Asian species *Vitis amurensis* [18,34,35]. Thanks to recent advances in the sequencing and annotation of the *P. viticola* genome, the first Avr locus in this pathogen was identified in a genome-wide association study (GWAS) [36]. Virulence towards *Rpv3.1* is associated with the absence of two secreted DEER proteins, and the locus presents an important allelic diversity. The interaction between *P. viticola* and *Rpv3.1*-carrying plants thus fits a gene-for-gene relationship, in which resistance is mediated by the recognition of specific pathogen effector(s) [37].

This raises the question of whether the newly observed virulences in *P. viticola* also result from the loss of effector genes, allowing escape from host immunity. Understanding the evolutionary trajectories of pathogen populations is also crucial to inform the deployment and management of new resistant grapevine varieties. It remains to be determined if virulences arise from the selection and subsequent spread of a single haplotype carrying a favorable variant, or rather from independent mutations at the same loci in different subpopulations.

In this study, we aimed to characterize in parallel the genetic bases of the adaptation to two major grapevine resistances in *P. viticola*. Using quantitative trait locus (QTL) mapping in two F1 populations, we identify the genomic regions linked to virulence towards *Rpv12* and *Rpv10*. We show that both virulences are determined by dynamic effector-rich regions, yet their modes of inheritance differ, the first being recessive and the second dominant. The structure of *P. viticola* populations suggests that *Rpv12* breakdown has occurred independently across multiple winegrowing regions, while virulence towards *Rpv10* was likely acquired through a single admixture event. These results highlight the diverse evolutionary pathways through which a specialized pathogen can adapt to plant resistances.

## Materials and methods

### Pathogen strains

The generation of the two F1 mapping populations was described in [38]. The first cross involved strains Pv412_11 and Pv2543_1 (N = 162) with contrasting pathotypes on Rpv3.1 and Rpv12 hosts (Table 1) [35]. The second progeny was the result of a cross between Pv412_11 and Pv1419_1 (N = 189), the latter being able to overcome *Rpv10* (Table 1). For brevity, these two F1 populations are hereafter referred to as 412x2543 and 412x1419.

**Table 1. Origin and pathotype of parent strains.**

| | | Pathotype on | | |
|---|---|---|---|---|
| | Origin | Rpv3.1 | Rpv10 | Rpv12 |
| **Pv412_11** | Canton of Ticino, Switzerland | virulent | avirulent | avirulent |
| **Pv2543_1** | Baranya county, Hungary | avirulent | avirulent | virulent |
| **Pv1419_1** | State of Baden-Württemberg, Germany | virulent | virulent | avirulent |

In addition, a backcross population was produced in order to further test the dominance of virulence towards *Rpv10*. The virulent strain cPv44_1 from the 412x1419 population was backcrossed to the parent Pv1419_1. Crossing, maturation and oospore retrieval were performed as described in [38].

The progeny is referred to as the BC-1419 population (N = 51).

## Phenotyping experiments

**Plant material.** *P. viticola* strains were phenotyped on a set of grapevine cultivars carrying different Rpv factors. We used cv. 'Cabernet-Sauvignon' (susceptible reference), cv. 'Regent' (*Rpv3.1*), cv. 'Muscaris' (*Rpv10*) and cv. 'Fleurtai' (*Rpv12*). The BC-1419 population was also phenotyped on an additional cultivar called 'Solaris' that carries *Rpv10*.

**Plant inoculation.** F1 progenies were inoculated on leaf discs of susceptible and resistant plants in separate experiments. Due to the large number of strains to phenotype, progenies were divided in half and inoculated in two parts, ten days apart, with a set of reference strains repeated in all experiments. Grapevine scions grafted onto the *Vitis berlandieri x riparia* 'SO4' rootstock were grown in a greenhouse without chemical treatment and under natural photoperiod conditions for 6 to 7 weeks. Leaf discs preparation, inoculation and incubation were performed as described previously [35]. Briefly, strains were initially propagated on detached leaves of cv. 'Cabernet-Sauvignon'. One day before the experiment, infected leaves were gently rinsed with distilled water to ensure the production of fresh sporangia for the inoculation the next day. Sporangia from each strain were suspended in sterile water and concentrations were adjusted to $10^5$ sporangia per ml using a Scepter 2.0 portable particle counter (Millipore). Leaf discs were excised from the fourth leaf below the apex, and placed on wet filter paper in a 12x12 cm Petri dish. Each suspension was sprayed on one Petri dish containing 5 discs per variety. For a given interaction, each disc came from a different plant. Plates were sealed with plastic film, and then incubated for 6 days in a growth chamber at 18°C with a 12:12 photoperiod. The BC-1419 population was phenotyped with the same method by inoculating 8 discs of cv. 'Cabernet-Sauvignon', 'Muscaris' and 'Solaris', and 4 discs of 'Fleurtai'.

**Traits measurement.** The percentage of sporulation area was calculated on high-resolution pictures taken at 6 days post-inoculation (dpi) using an in-house image analysis program (code available on https://gitlab.com/grapevinedownymildew/notebook_image_analysis). On the same pictures, necrotic lesions were visually assessed using an ordinal scale based on their size and appearance. Discs received a score between 1 (large non-specific necrotic speckles) and 5 (small dark lesions indicative of an efficient HR), with complete absence of necrosis scored as 0 (S1 Fig).

## Statistical analyses

Analyses were conducted using R v4.1.3. Broad-sense heritabilities (H2) were calculated for each trait on the different inoculated hosts using the function H2cal implemented in r/inti v0.6.6 [39]. As F1 offspring of the same population were phenotyped in two parts, we fitted a linear mixed model with the experiment (Exp) treated as a random intercept effect, and the percentage of sporulation area (Spo) for each individual (Ind) were calculated as best linear unbiased estimations using r/emmeans v1.10.2 [40]. This was done separately for each inoculated variety. The model was noted as $Spo_{ijk} = \mu + Ind_i + u(Exp)_j + \varepsilon_{ij}$ with $u(Exp)_j \sim \mathcal{N}(0, \sigma^2_{Exp})$ and $\varepsilon_{ijk} \sim \mathcal{N}(0, \sigma^2)$.

For the study of the BC-1419 population, we tested the effect of the genotype (Geno), the inoculated host (Host), and their interaction on the sporulation area (Spo) by fitting a linear mixed model with the Ind variable treated as a random intercept effect. The model was noted as $Spo_{ijkl} = \mu + Geno_i + Host_j + (Geno : Host)_{ij} + u(Ind)_k + \varepsilon_{ijkl}$ with $u(Ind)_k \sim \mathcal{N}(0, \sigma^2_{Ind})$ and $\varepsilon_{ijkl} \sim \mathcal{N}(0, \sigma^2)$. Pairwise comparisons between groups were performed with r/emmeans.

## QTL mapping

We employed a pseudo-testcross mapping strategy, which is suitable for a self-incompatible and highly heterozygous species such as *P. viticola*. Parental linkage maps were built based on targeted genotyping-by-sequencing of SNP markers and were previously presented in [38]. Triploid strains were not included in the following analyses. Thus, 6 individuals were removed in the 412x1419 progeny and 7 in 412x2543, resulting in respectively 183 and 155 genotypes effectively used.

Single interval QTL mapping was performed using the scanone function implemented in r/qtl v1.60 [41]. A square root transformation was applied to measures of sporulation area to normalize their distribution. LOD scores were calculated using a normal model and Haley-Knott regression. Significance levels were computed with 1000 genome scan permutations. QTL boundaries were determined by calculating credible intervals with the bayesint function ($\alpha = 0.05$). The percentage of variance explained by each QTL was calculated using the fitqtl command. Composite interval mapping was also tested using the cim function with different window sizes and markers set as covariables. This did not reveal additional QTLs and did not change credible intervals.

## Analysis of putative effector genes in the QTLs

The following analyses were conducted using the most recent assembly of reference strain Pv221_1 [36]. Genes located in QTL intervals were checked for the presence of signal peptides (SP), RXLR and/or dEER motifs, as well as LWY domains, a structural fold associated with oomycete effectors. Secreted proteins were predicted using SignalP 5.0 [42]. LWY domains were searched with HMMER 3.2 (hmmer.org) as described in Dussert et al. [12], using the HMM profile from Boutemy et al. [25].

Structural homology searches were conducted with Phyre2 [43]. Predictions of protein structure were performed using AlphaFold2 [44] as implemented in ColabFold v1.5.2 [45] with default settings. Visualization and superimposition of protein structures were visualized with UCSF ChimeraX v1.5 [46]. The experimental structure of the *Phytophthora sojae* PSR2 effector was retrieved from the Protein Data Bank (https://www.ebi.ac.uk/pdbe/pdbekb/proteins/E0W4V5).

Expression of the genes present in the identified QTLs was checked during plant infection by the reference avirulent strain Pv221_1. Transcript analysis was conducted using processed RNA-sequencing data available from Dussert et al. [12] (SRA BioProject PRJNA329579).

## Haplotype-resolved assembly of parent strain Pv1419_1

We benefited from a *de novo* assembly of the parent strain Pv1419_1. High-molecular weight DNA was extracted from sporangial tissues using the same protocol described in Dussert et al. [12]. DNA fragments were sequenced on a PacBio Sequel II system, yielding long high-fidelity reads. They were assembled into two sets of 17 pseudochromosomes using the haplotype-aware pipeline HaploSync [47]. Assembly procedure, quality assessment and annotation are detailed in S1 Methods.

## Genotyping of the backcross population by amplicon length polymorphism

After the detection of a QTL in the 412x1419 F1 progeny, we aimed to follow the inheritance of virulence in the BC-1419 population. Indels were identified between the haplotypes carried by Pv1419_1 at the QTL: one at each edge of the

physical interval, and one co-segregating with the QTL peak. For each genotype, different profiles of amplicon lengths were generated by PCR and visualized by electrophoresis on agarose gels (S1 Methods). Primer sequences and annealing temperatures are indicated in S2 Table.

**Whole genome sequencing of additional strains**

We sequenced 41 new strains, all belonging to *P. viticola f. sp. aestivalis*, the only species present worldwide [10]. Additionally, whole genome sequencing (WGS) data were obtained for six F1 individuals and one BC-1419 strain, which enabled the phasing of inherited variants in the QTLs we detected. DNA was extracted from sporangial tissues following a CTAB protocol adapted from Möller et al. [48] and previously detailed in [38]. Libraries were prepared using an Illumina DNA TruSeq kit. Sequencing was performed at the GeT-PlaGe facility (Toulouse, France) with a NovaSeq6000 to produce 2x150 bp paired-end reads. This applied to all samples except Canadian isolates, for which DNA was sequenced by Beckman Coulter Genomics (Grenoble, France) on an Illumina HiSeq 2000 sequencer (2x100 bp paired-end reads).

**Population structure analysis**

We constructed a panel of sequences that included strains of various pathotypes collected from different European regions as well as other continents (S1 Data). All samples that have a number as a suffix ('_1' or '_11') were derived from monosporangium isolation as described in Paineau et al. [35]. Short DNA reads were obtained either in previous studies or for the present one (S1 Data). In total, 56 wild strains and 7 F1 or BC-1419 individuals were included.

The population structure was investigated using SNP data from wild strains. Read mapping, variant calling and filtration are detailed in S1 Methods. Variants were pruned to retain those in approximate linkage equilibrium with PLINK 1.9 [49]. The parameters used were a window size of 50 variants with a shift of 10 variants at each step, and a $r^2$ threshold of 0.1. After pruning, 89,673 SNPs remained. Using PLINK, KING-robust kinship coefficients were computed to check for closely related samples. A Principal Component Analysis (PCA) was performed using r/adegenet v2.1.10 [50]. Population clusters and individual ancestries were then inferred using ADMIXTURE v1.3 with unsupervised analysis [51]. The program was run for K = 1 to K = 5 with 200 bootstrap replicates each time, and five-fold cross-validation (CV) errors were calculated for each K.

Ancestry segments in admixed individuals were identified using MOSAIC v1.5.1 [52]. As this approach required variant phasing, we used only a subset of SNPs that were fixed in each "ancestral" population (homozygous reference allele in the subset of European strains not showing admixture with the North American cluster, homozygous alternate allele in the North American population). Thus, 328,439 potential SNPs were suitable for the analysis, including 33,832 in chromosome 16, where the region of interest was located. The set of variants in this chromosome was filtered as described above, and then thinned with PLINK to achieve a density of about 1 SNP/kb, retaining 3876 SNPs. Genotypes of admixed strains were then phased using Beagle v5.2 [53] with the two mentioned sub-populations set as references. Standard parameters were used, except for the sliding windows length that was set to 1 Mb with 0.1 Mb overlaps. Local ancestry along the genome was finally estimated using MOSAIC with default parameters and rephasing enabled.

**Analysis of the *AvrRpv12* locus**

A maximum-likelihood phylogenetic tree was built using protein sequences of putative effectors in the locus to put it in relation with the physical distance between genes. We used the method implemented in MEGA11 with an initial Neighbor-Joining tree, 1000 bootstrap replicates and default parameters (JonesTaylor-Thornton substitution model, uniform rates among sites, fast SPR heuristic method).

The presence-absence of genes was inferred from the depth of coverage at the locus. Per-base read depth was obtained with samtools v1.18 [54]. The mean coverage value was calculated on 5 kb windows and normalized based on the genome-wide average for each strain, assuming diploidy.

PLOS Pathogens

In addition, we investigated signs of a selective sweep at the locus in virulent strains. The analyses were carried out at the chromosome scale by ordering and orienting contigs based on the linkage map. Runs of homozygosity were detected and visualized with r/detectRUNS v.0.9.6 [55] using sliding windows of 15 SNPs. The parameters required to identify a segment were set to a minimum number of 20 SNPs, a maximum number of 1 heterozygous sites, a minimum length of 50 kb, and a maximum gap length of 100 kb.

We also compared signals of positive selection in virulent versus avirulent strains from Hungary (n = 15). To do so, we calculated the cross-population Extended Haplotype Homozygosity (XP-EHH), which detects SNPs approaching fixation in a population while remaining polymorphic in the other [56].

We applied the method implemented in selscan v2.0.3 [57] for unphased data, with default parameters except for a maximum gap length of 250 kb.

### Detection of a S-AvrRpv10 candidate effector gene in North American samples

We assessed the presence of S10-SP6, one the candidate effector genes associated with virulence towards *Rpv10*, in North American *P. viticola* populations. To do so, we tested 93 DNA samples of *P. viticola f. sp. aestivalis*, which were collected from wild and cultivated hosts in various locations in the United States for previous studies [10,58]. A duplex qPCR assay was designed, including the target gene as well as the PvTUB gene as a positive amplification control. Specific TaqMan fluorogenic probes were manufactured by Eurogentec (Seraing, Belgium). Duplex assays were performed using a QuantStudio 5 Real-Time PCR System (Applied Biosystems) with technical triplicates. The reaction mix was composed of 10 µL of Invitrogen Platinum II Hot-Start (2X), primers for each gene (0.5 µM), probes (0.2 µM) and 1 µL of genomic DNA for a final reaction volume of 17.5 µL. The cycling parameters were 3 min at 95°C followed by 40 cycles of 10 s at 95°C and 45 s at 60°C. Primer and probe sequences are indicated in S3 Table. The S10-SP6 gene was considered present in a sample if the cycle threshold was inferior to 35 and PvTUB was successfully amplified.

## Results

### Mapping of three major QTLs involved in resistance breakdown

Two biparental F1 populations were derived from the cross of *P. viticola* strains exhibiting contrasting pathotypes on three grapevine resistance factors (Table 1). Phenotyping on resistant cultivars revealed segregation on *Rpv3.1* and *Rpv12* plants in the 412x2543 progeny, and segregation on *Rpv10* plants in the 412x1419 progeny. Due to the pseudo-testcross mapping strategy, two parental linkage maps were built for each cross and QTLs could be detected when a parent strain was heterozygous at the locus. One major QTL was detected for each R gene, in different parental maps (S2 Fig).

We first confirmed the position of the *AvrRpv3.1* locus, which was previously identified by GWAS [36]. In the 412x2543 population, 88 out of 162 individuals did not trigger HR on cv. 'Regent' (average necrosis score = 0), suggesting that this virulence towards *Rpv3.1* segregated with a 1:1 ratio (Chi-squared test, p = 0.31) (S3A Fig). A single major QTL was detected at the edge of LG11 in the linkage map of the avirulent parent Pv2543_1 (S3B and S3C Fig), with a credible interval of 5.1 cM. The QTL explained 46.1% of the variation in sporulation area and 87.1% of the variation in necrosis score.

The genetic interval corresponded to a 148-kb long physical segment encompassing two effector genes, Primary_000014F.g164 and Primary_000014F.g165 (P14g164 and P14g165). Read depth analysis showed that P14g164 and P14g165 were partially or totally deleted in both haplotypes of the virulent parent Pv412_11 (S3D Fig). These genes were present only in the avirulent haplotype of Pv2543_1, which is consistent with their role as Avr factors recognized by Rpv3.1 [36].

### Virulence towards *Rpv12* is associated with the loss of RXLR genes

The 412x2543 F1 population also segregated with respect to the interaction with *Rpv12*. The sporulation area varied along a continuum from 0 to 11% (Fig 1A). Trait heritabilities were lower than those observed in the interaction with

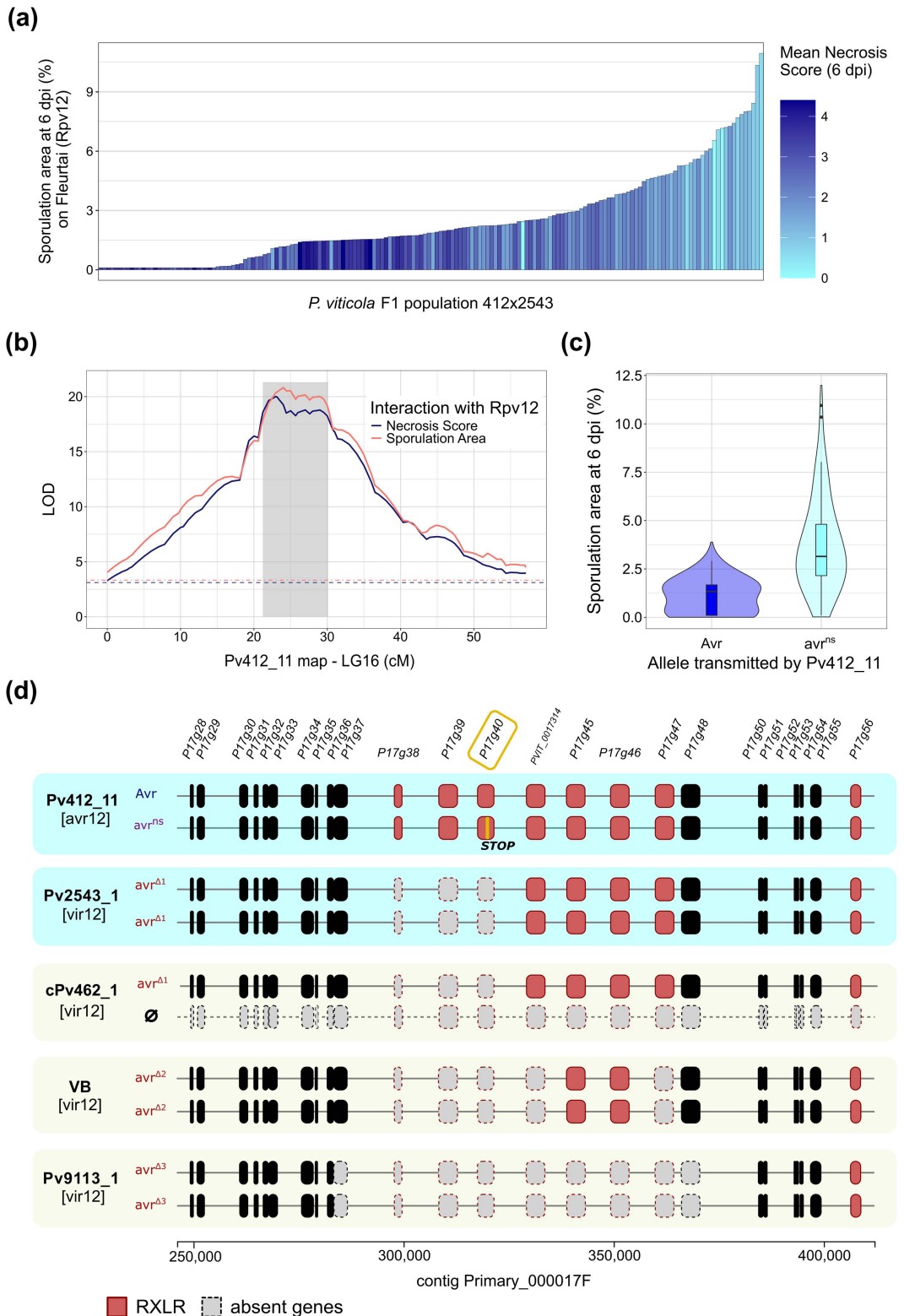

**Fig 1. Mapping and characterization of the *AvrRpv12* locus in a biparental population of *P. viticola*.** **(a)** Phenotypes distribution in the 412x2543 F1 progeny (N = 162) on cv.'Fleurtai' (*Rpv12*). **(b)** QTL mapping of *Rpv12*-breakdown in the linkage map of the avirulent parent Pv412_11. The gray

area indicates the credible interval of the QTL. Dashed lines indicate the LOD significance thresholds determined using 1000 permutations. Results on other linkage groups and other cultivars are available in S2 Fig. **(c)** Distribution of the sporulation area on *Rpv12* depending on the marker allele at the QTL peak. Horizontal lines in the boxplots signal the 25th, 50th and 75th percentiles. **(d)** Allelic configurations of parent strains (blue boxes) and other virulent strains (beige boxes) in the physical interval of the QTL. For each strain, the two haplotypes are represented. Genes are colored in black or red if they possess a RXLR motif. Dashed gray boxes represent absent genes. Deletion patterns were deduced from read depth (S5 Fig). In the Pv412_11 haplotype associated with virulence (avr^NS), an orange stroke signals the 1-nt indel causing a premature stop codon in P17g40. Strain cPv462_1 is an aneuploid offspring that lacks the copy of chromosome 16 transmitted by Pv412_11.

*Rpv3.1*, with H2 = 0.40 for the sporulation area and 0.35 for the necrosis score. High levels of sporulation tended to be associated with light necrosis surrounding sporulation spots (average necrosis score < 3) (Fig 1A). This did not correspond exactly to the parental phenotype, as Pv2543_1 infected *Rpv12* plants without triggering HR.

A single major QTL was detected in LG16 in the linkage map of the avirulent parent Pv412_11, with a similar profile for the two measured traits and a genetic interval of 8.9 cM (Fig 1B). No QTL was detected on *Rpv12* plants in the linkage map of the virulent parent (S2 Fig). The variance explained by the Pv412_11 QTL was 28.9% for the sporulation area and 31.7% for the necrosis score. The distributions of sporulation area between the two alleles partly overlapped (Fig 1C). As Pv412_11 itself showed a quasi-absence of sporulation, these two alleles were seemingly revealed by the cross with the virulent strain.

We analyzed the physical region underlying the QTL which corresponded to a 157-kb long segment in the Pv221_1 reference genome (Fig 1D). This interval comprises 25 coding sequences. One of them was incorrectly annotated so we used the name given in the first version of the *P. viticola* genome annotation instead (PVIT_0017314) [12]. Most genes have no functional annotation (S2 Data). However, eight of them encode proteins that contain a secretion peptide (amino acids 1–20) and a RXLR motif (aa 49–52). Thus, we focused primarily on these genes as they are likely to encode cytoplasmic effectors. Transcriptome analysis showed that they were all expressed during plant infection by the avirulent reference strain Pv221_1. Their protein sequences vary from 608 to 1481 aa. All these RLXR genes are highly similar (>50% of amino acid identity) and belong to a family of 18 genes located in a 300 kb segment around the QTL (S4 Fig). Phylogenetic analysis showed that related sequences tend to be physically close (S4 Fig). Two genes, PVIT_0017314 and P17g45, have an increased read depth in both parents which is probably due to additional copies compared to the reference genome (S5 Fig). These elements suggest that this region is subjected to repeated tandem duplications generating copy number variation (CNV) and paralogous sequences.

Read depth analysis in the virulent parent strain Pv2543_1 revealed a large homozygous deletion containing three RXLR genes: P17g38, P17g39 and P17g40 (Figs 1D and S5). They code for proteins sharing structural homology with known oomycete effectors, such as PSR2 from *P. sojae* and RXLR12 from *Phytophthora capsici* (best hits obtained with Phyre2). HMM search also highlighted the presence of repeated LWY folds in the three proteins (respectively 3, 10 and 9 modules). We modeled the structure of P17g40 using AlphaFold2 and divided it based on the identified LWY modules, which revealed a clear and complete overlap with the PsPSR2 modules (S6 Fig).

Due to their absence in Pv2543_1, the 412x2543 offspring inherited only one copy of these genes. The other parent, Pv412_11, presented an important heterozygosity in their coding sequences, with respectively 9, 45 and 13 non-synonymous mutations. Whole-genome sequencing of three 412x2543 offspring enabled us to phase variants in these genes. P17g38 and P17g39 share respectively 98.5% and 98.4% nucleotide sequence identity between the two alleles, while P17g40 sequences are 99.2% identical. However, for this last gene, a 1-nt deletion is present in the allele associated with higher sporulation. This frameshift mutation leads to a truncated protein at the position 795/1298 (Fig 1D). We noted the haplotype avr^ns because of this nonsense mutation in a putative effector.

Given the major variations affecting RXLR genes in this locus, we hypothesized than one or several of them could correspond to an Avr gene recognized by *Rpv12*. We checked their status in other *P. viticola* strains. We found that all strains virulent towards *Rpv12* were affected by large deletions in the locus, but their lengths varied depending on the

location of origin (Figs 1D and S5). Hungarian strains presented the same deletion as Pv2543_1 (allele noted avr$^{\Delta 1}$). In the Swiss VB strain described in Wingerter et al. [18], 5 RXLR genes were absent (avr$^{\Delta 2}$). Finally, virulent Italian strains (like Pv9113_1) lacked 7 out of the 8 RXLR genes in the QTL (avr$^{\Delta 3}$). We also identified one Hungarian strain (Pv2963) that was fully avirulent and appeared hemizygous at the *AvrRpv12* locus (S5 Fig). This observation is consistent with a recessive virulent allele.

Compared to avirulent strains from similar geographical origins, *Rpv12*-breaking strains consistently exhibited long runs of homozygosity around the deletions (S7A Fig), which could be the result of a recent selective sweep. Besides, in our panel of Hungarian strains (n = 15), the strongest signal of positive selection was observed around the locus in the virulent subpopulation (S7B Fig).

Lastly, one genotype in the 412x2543 F1 progeny provided additional clues that the QTL corresponded to *AvrRpv12*. Strain cPv462_1 is aneuploid [38] and lacks the copy of chromosome 16 normally transmitted by the avirulent parent Pv412_11 (Fig 1D). This strain induced poor sporulation on all cultivars, possibly due to its karyotypic anomaly. Interestingly, it did not trigger any necrosis on *Rpv12* plants (average necrosis score = 0). The absence of HR was further verified by inoculating additional 'Fleurtai' leaf discs by droplet and observing the discs over one week. This confirmed that avirulence towards *Rpv12* is entirely carried by chromosome 16. Together, these findings led us to consider RXLR genes in the QTL as strong *AvrRpv12* candidates.

## A major QTL determines the partial breakdown of *Rpv10*

The 412x1419 F1 population (N = 189) was used for QTL detection in the interaction with *Rpv10*. A large part of the off-spring showed little to no sporulation on cv. 'Muscaris' (Fig 2A). The median sporulation area was only 0.5%, with the most aggressive strains reaching 6% of sporulation area. For comparison, the median sporulation area reached 7.2% on the susceptible cultivar 'Cabernet-Sauvignon'. Necrotic lesions were present on almost every leaf discs (Fig 2A). Sporulation area and necrosis score showed a heritability of 0.55 and 0.59 respectively.

A single major QTL was detected on LG16 in the linkage map of the virulent parent Pv1419_1 (Fig 2B). The QTL could be narrowed down to a genetic interval of 2.4 cM. It explained 27.5% of the variance in sporulation area and 35.4% of the variance in necrosis score. No QTL was found on other grapevine cultivars or in the other parental map (S2 Fig). In our setting, a QTL could be detected if the parent strain were heterozygous, which suggested that the virulence of Pv1419_1 was due at least in part to a dominant or co-dominant locus. Because the HR was still activated in cv. 'Muscaris', we refer to the observed phenotype as a partial breakdown of *Rpv10*.

We then explored the physical segment underlying the identified QTL. It was found on the same linkage group as the *AvrRpv12* locus but the two regions are clearly distinct: the two intervals are located on both sides of the centromere [38] and separated by at least 1 Mb. Despite the limited genetic interval, the physical segment was large: at least 537 kb, spread over two contigs (Primary_000044F and Primary_000062F). This was mainly due to the absence of recombination in a 248-kb long region which coincided with the peak of the QTL. A large fraction of the locus length (24%) was annotated as repeats, mostly corresponding to Copia-like long terminal repeat TEs.

A striking feature of the QTL interval was a strong enrichment in genes coding for secreted proteins. It contains 120 genes, 50 of which encode an N-terminal SP. This represents 42% of the coding sequences in the interval, compared to 9.7% in the entire genome. Given the large size of the region, we focused primarily on these genes because of their potential role in the interaction with the host. All genes except one were expressed during plant infection by the reference avirulent strain Pv221_1 (S3 Data). Almost all of them code for proteins of similar lengths (300–400 aa) with typical oomycete effector features. They lack the RXLR motif, but possess an N-terminal DEER sequence (typically in aa positions 55–58) and exhibit repeated LWY domains (1–4, mostly 3, S3 Data). Therefore, the QTL constitutes a hotspot of putative effectors that could be involved in the partial breakdown of *Rpv10*. However, we were still limited in our comprehension of the genomic region due to the lack of an assembly of the distinct haplotypes.

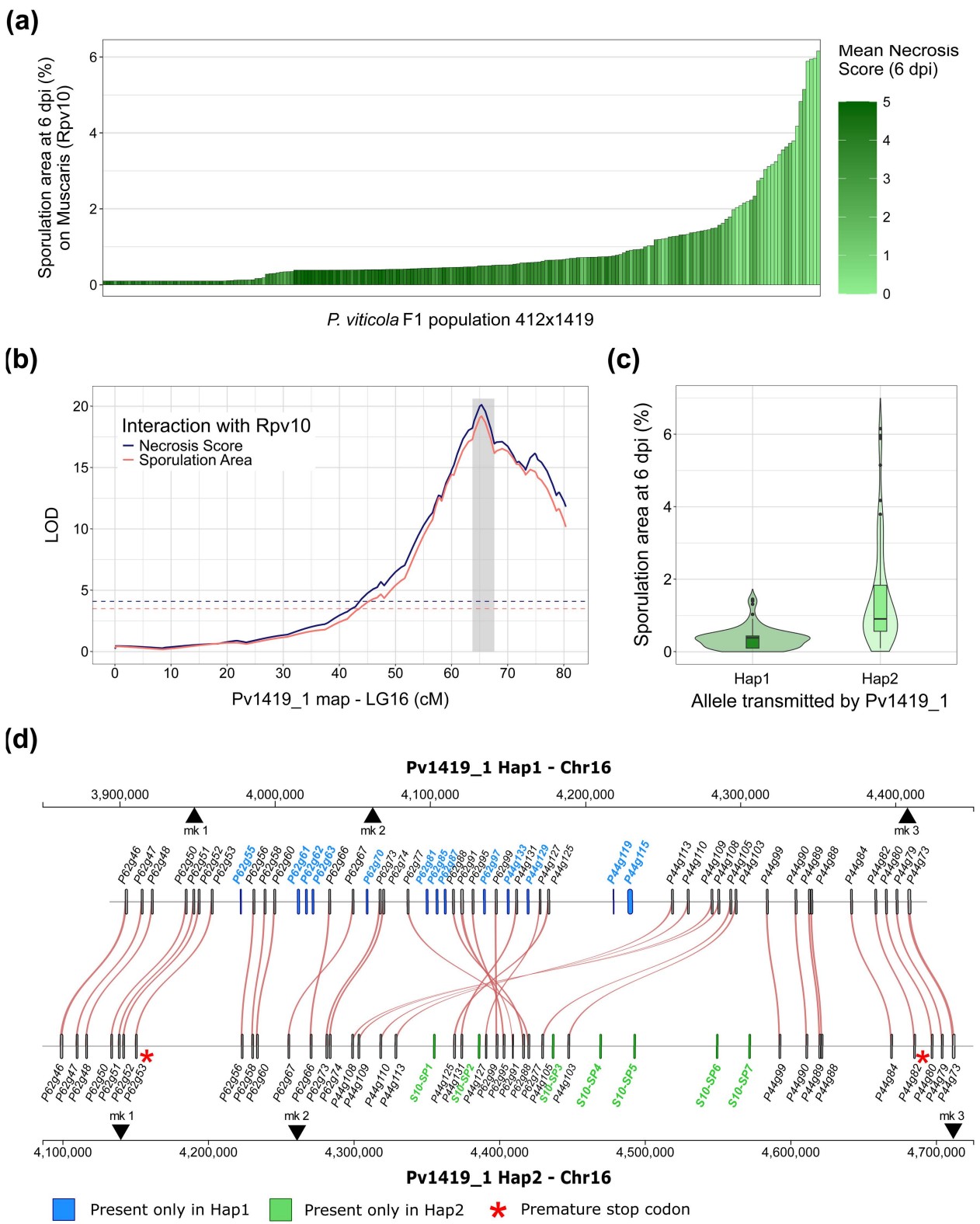

**Fig 2. Identification of a locus determining the partial breakdown of Rpv10 in a biparental population of *P. viticola* (a) Phenotypes distribution in the 412x1419 F1 progeny (N = 189) on cv.'Muscaris' (*Rpv10*). (b)** QTL mapping of *Rpv10* breakdown in the linkage map of the virulent parent

Pv1419_1. The gray area indicates the credible interval of the QTL. Dashed lines indicate the LOD significance thresholds determined using 1000 permutations. Results on other linkage groups and other cultivars are available in S2 Fig. **(c)** Distribution of the sporulation area on *Rpv10* depending on the marker allele at the QTL peak. Horizontal lines in the boxplots signal the 25th, 50th and 75th percentiles. **(d)** Comparison of the two haplotypes of Pv1419_1. The avirulent haplotype is represented at the top (Hap1). The haplotype associated with partial breakdown of *Rpv10* is at the bottom (Hap2). Genes encoding putative secreted proteins are represented and linked in red when they are present in both haplotypes. Sequences present only in one haplotype are colored in blue (Hap1) or green (Hap2). Genes are named after their annotation in the reference assembly of Pv221_1. Coding sequences of Hap2 that are absent from the reference genome were named S10-Secreted Protein (SP) 1 to 7. Red asterisks signal genes containing premature stop codons. Black triangles indicate the positions amplified by PCR for the genotyping of the BC-1419 backcross progeny (S2 Table).

### Extensive structural variation between the two parental haplotypes

To investigate differences between haplotypes, HiFi reads of strain Pv1419_1 were used to build a chromosome-scale assembly (2n = 34) in which both haplotypes were fully represented. We used the SNPs included in the linkage map to associate haplotypes with their corresponding phenotypes and verify that the phasing was correct along the QTL.

The two parental haplotypes carried by Pv1419_1 presented extensive structural variation (Fig 2D). The first haplotype (Hap1) corresponded to the fully avirulent phenotype. It stretched over 507 kb, and its gene content and order were identical to the reference strain Pv221_1. By contrast, the second haplotype (Hap2) was highly divergent and considerably larger (617 kb). The central part of the segment presented a large inversion that affected around 213 kb in Hap1 and 122 kb in Hap2. The position of this structural variation coincided with the non-recombining region in the linkage map. This large-scale inversion probably explains why crossovers were prevented in such a long interval. The gene content also differed: 13 secreted protein genes are missing in Hap2 and were not found anywhere else in the chromosome (Fig 2D). Interestingly, this is counter-balanced by the presence of 7 new genes that are exclusive to Hap2 (genes labeled SP1 to SP7), four of which are located in a 100-kb insertion located near the inversion. All seven genes code for secreted proteins that possess an N-terminal DEER motif and 3 repeated LWY domains (S4 Data). They correspond to additional copies or paralogous sequences of neighboring genes (between 59 and 94% of aa identity) (S4 Data).

We also examined point mutations affecting genes conserved in both haplotypes. Two genes, P44g82 and P62g53, were affected by premature stop codons in Hap2 (red asterisks in Fig 2D). Between the two haplotypes, protein sequences were 97.3% identical on average (min. 93.5%, max. 100%) (S4 Data). Altogether, the haplotype responsible for the partial breakdown carried both large-scale structural variants and numerous nucleotide substitutions.

### Partial virulence towards *Rpv10* is dominant in a backcross population

By contrast with the other QTLs, the one involved in *Rpv10* breakdown was not recessive. Thus, we provisionally designate the locus Suppressor of Avirulence towards Rpv10 (*S-AvrRpv10*). For brevity, we labeled the allele associated with full avirulence "s10" and the active allele "S10". The 412x1419 F1 population was only composed of S10/s10 and s10/s10 individuals. Thus, the characterization of the homozygous S10/S10 genotype on *Rpv10* was crucial to determine if the S10 allele was fully dominant or co-dominant.

We generated and phenotyped the backcross population BC-1419 (N = 51). Strains were genotyped by PCR-based amplicon length polymorphism (S5 Data). Three markers were designed, with one cosegregating with the QTL peak (position "mk2" on Fig 2D). The observed segregation was in accordance with the expected 1:2:1 ratio (18:23:11, p = 0.22).

The genotype at the *S-AvrRpv10* locus had no effect on the susceptible cultivar 'Cabernet-Sauvignon' (Fig 3). Conversely, all strains were fully avirulent towards *Rpv12*. On *Rpv10* cultivars 'Muscaris' and 'Solaris', the sporulation area was higher in strains carrying one or two copies of the S10 allele. However, there was no difference of aggressiveness between heterozygous S10/s10 and homozygous S10/S10 strains (Fig 3). Weak necrosis was still observed on discs inoculated with S10/S10 individuals (average necrosis score = 2.30 on 'Muscaris' and 2.15 on 'Solaris'). Therefore, we conclude that the S10 allele is

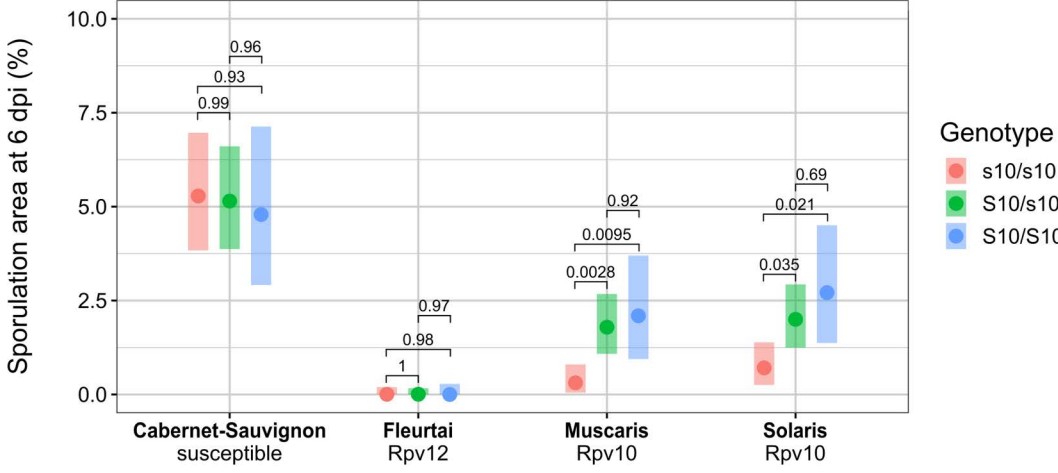

**Fig 3. Phenotyping of a *P. viticola* backcross population segregating for partial virulence towards *Rpv10*.** The BC1419 population (N=51) was obtained by crossing S10/s10 strains, thus generating three genotypes at the QTL in the progeny. Offspring were genotyped using three markers by PCR-based amplicon length polymorphism (S2 Table). The genotypes shown were obtained using a marker co-segregating with the peak of the QTL in the F1 population (Fig 2D). Dots indicate the adjusted means and colored bars represent confidence intervals (α=0.05). Tukey's adjusted p-values were calculated for pairwise comparisons with r/emmeans [40]. Full genotyping results are available in S5 Data.

fully dominant and that its effect is specific to the interaction with *Rpv10*. It is however not sufficient to completely prevent the immune response in *Rpv10* cultivars, as indicated by the presence of necrosis for all genotypes.

## Population structure points to different scenarios of adaptation to *Rpv* genes

We took advantage of WGS data obtained from a panel of strains of various origins and pathotypes (S1 Data) to explore the population structure of *P. viticola* in light of our results above. In particular, we aimed to elucidate if the adaptation to each resistance factor in Europe stemmed from a common genetic background.

First, we performed a PCA based on a set of 89,673 SNPs (Fig 4A). The first PC separated North American strains from the rest of the world while the second PC mostly corresponded to an east-west gradient in Europe. Genetic clustering with the ADMIXTURE program was in accordance with the geographical structuration suggested by PCA (Fig 4B). The CV error was lowest with three clusters (K=3), making it the best-fitting model (S8 Fig). American strains clustered together consistently (cluster 1). With K=3, French and Spanish strains tended to be included in cluster 2 while cluster 3 regrouped most Italian and Hungarian strains.

According to this population structure analysis, *Rpv12*-breaking strains are genetically similar to avirulent strains collected near them and do not form a distinct genetic group (Fig 4). Thus, virulent strains do not belong to a unique lineage but rather emerged from local *P. viticola* populations. In addition, they displayed different lengths of deletion at the *Avr-Rpv12* locus (avr$^{Δ1-3}$, Fig 1D), which hints at the selection of independent mutation events in each location.

All strains with high aggressiveness on *Rpv10* were collected in the Upper Rhine Plain, a winegrowing region located around the French-German border. Intriguingly, they showed more genetic similarity to North American strains than the rest of the panel, as seen along the first PC (Fig 4A). In particular, Pv1419_1, in which the *S-AvrRpv10* locus was identified (Fig 2), was placed halfway between the North American group and avirulent strains from the Upper Rhine Plain. Genetic clustering revealed an admixture from the North American population in *Rpv10*-breaking strains, in variable proportions depending on the sample (17–57%) (Fig 4B). This signal was not observed in avirulent strains from the same

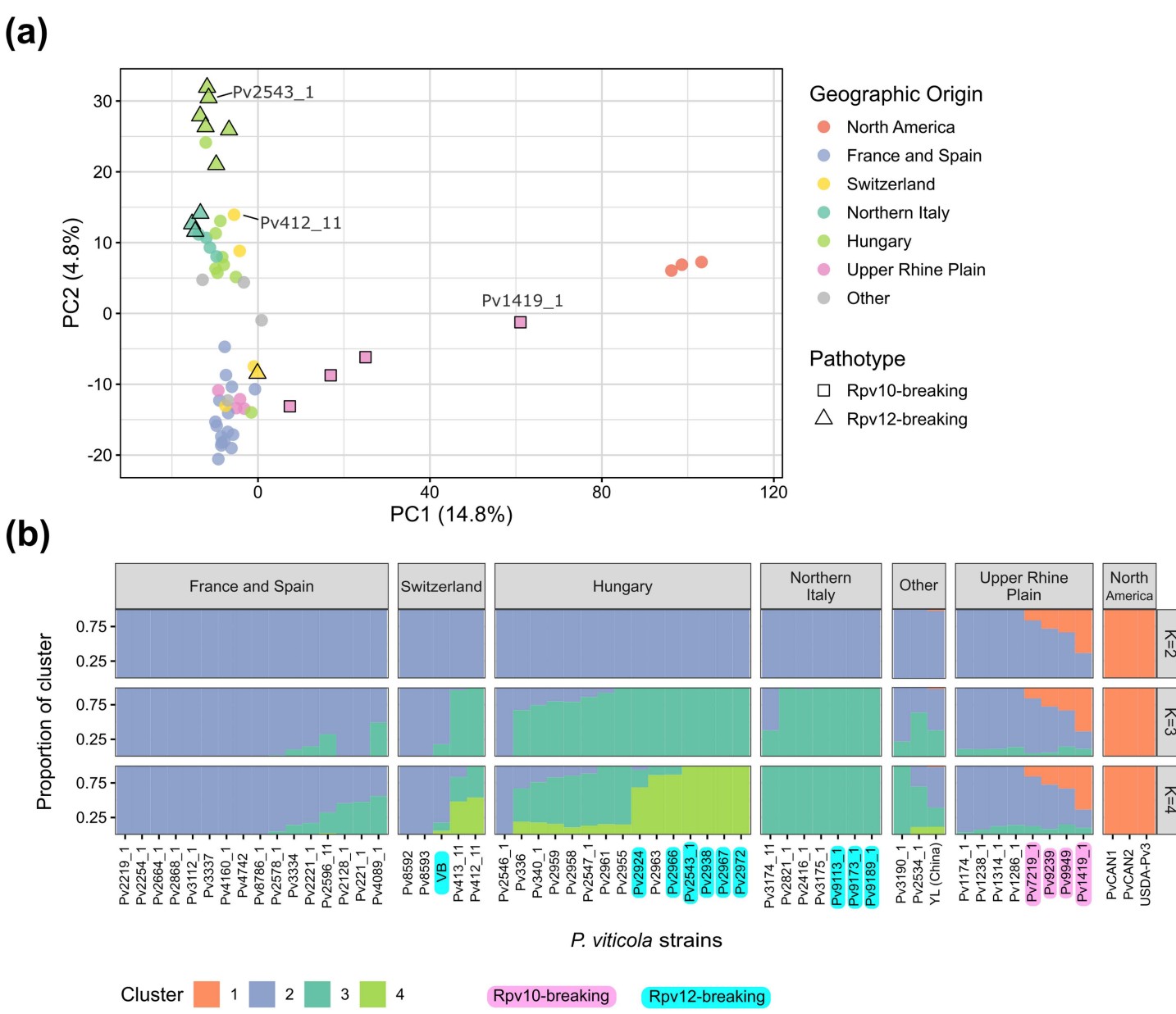

**Fig 4. Population structure of *P. viticola* strains with various origins and pathotypes.** Analyses were based on 89,673 SNPs in a panel of 57 strains. The Upper Rhine Plain is a winegrowing region at the French-German border. **(a)** Principal Component Analysis. The first two principal components (PC) are represented, with the percentage of variance explained indicated in parentheses. Strains are colored according to their geographical origin, and shapes indicate pathotypes of interest. The positions of parent strains used for QTL mapping are signaled. **(b)** Unsupervised genetic clustering performed with the ADMIXTURE program. Bar plots represent estimated genome ancestry fractions for each strain. Results are shown for 2 to 4 clusters (K). Cross-validation error was lowest with K = 3. Strains are separated by geographical origin. The "Other" category regroups non-European strains collected (from left to right) in Georgia, Lebanon and China. Strains overcoming *Rpv10* or *Rpv12* are highlighted in pink and cyan, respectively. Plot made with r/starmie v0.1.3.

region. The balanced ancestry of Pv1419_1 suggests that it could be an early generation hybrid between populations, in accordance with its very high heterozygosity rate (1.41 heterozygous sites per kb versus 0.70 on average in non-admixed strains). The lower admixture levels observed in strains collected more recently (between 2020 and 2023) could be the result of recurrent backcrossing with the local population.

### The *S-AvrRpv10* locus coincides with an admixed genomic region

To understand if virulence towards *Rpv10* was linked to a potential admixture, we analyzed the levels of each ancestry along the chromosome 16 in admixed strains. Ancestral segments were defined using a subset of 3876 fixed SNPs between the North American population and European strains for which no American admixture was detected (Fig 4B). Parent strain Pv1419_1 is heterozygous at the *S-AvrRpv10* locus (S10/s10) (Fig 2D) and it showed a mix of the two ancestries along the entire chromosome (Fig 5). Descendants of Pv1419_1 had different ancestry profiles depending on their genotype. For example, cPv44_1 (S10/s10) presented a mixed ancestry while cPv43_1 (s10/s10) had a full European ancestry. In addition, an homozygous S10/S10 individual from the BC-1419 population presented a full American ancestry (bcPv44–04, Fig 5). The profile of wild strain Pv7219_1 was fully American along the locus except for the distal end where it was mixed. This pattern was consistent with the markers used to genotype the BC-1419 population (S5 Data). Other *Rpv10*-breaking strains displayed either a mixed or a fully American ancestry.

In addition, to challenge the hypothesis of a North American origin, we tested 93 DNA samples from the USA for the presence of a candidate effector from the locus. We selected the candidate effector S10-SP6 as it was encoded by one of the most divergent genes and was absent from all the European strains in the panel. The gene was detected in 43% of

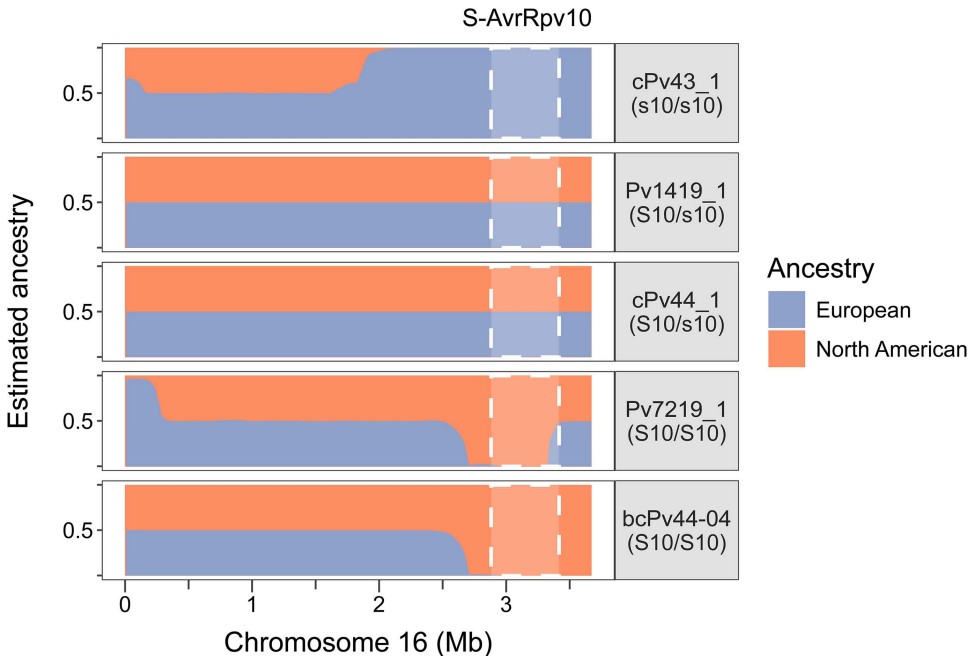

**Fig 5. Ancestry segments in admixed *P. viticola* strains.** Estimation of local ancestry was conducted on chromosome 16 using a subset of 3876 SNPs. Non-admixed European and North American samples were set as ancestral populations. Proportions of ancestry were estimated for natural admixed strains as well as descendants of Pv1419_1 (F1: cPv43_1 and cPv44_1, backcross: bcPv44-04). The *S-AvrRpv10* QTL limits (Fig 2) are indicated by white dashed lines. Five strains of different genotypes are shown: S10 indicates the dominant allele associated with partial virulence while s10 corresponds to the recessive avirulent allele.

the samples, distributed across different states and collected on wild as well as cultivated grapes (S6 Data). This confirms that one of the genes confined to the virulent haplotype is present in North American populations.

Thus, the avirulent allele s10 is associated with a standard European genetic background, while the partially virulent allele S10 corresponds to an admixed region. This suggests that the adaptation to *Rpv10* occured through a genetic contribution from an extra-European population, possibly originating from North America.

## Discussion

In this study, we successfully mapped two new loci in the *P. viticola* genome involved in the interaction with *Rpv10* and *Rpv12*, two key resistance factors of European grapevine breeding programs. Furthermore, sequencing a panel of strains representing the pathogen's diversity provided insights into the dynamics of virulence emergence, both at the genomic and population levels.

### Dynamic regions foster rapid adaptation to resistances

The three major QTLs involved in resistance breakdown share many common features. They contain clusters of genes encoding secreted proteins with typical oomycete effector motifs (RXLR for *AvrRpv12*, dEER for *AvrRpv3.1* and *S-AvrRpv10*). These proteins also display canonical or modified LWY domains, which participate in a modular structure. Closely related sequences tend to be physically close and the limit is often blurred between paralogous genes and additional copies. Analysis of different haplotypes confirmed that these loci are evolutionarily dynamic, with their effector repertoire varying between alleles. *AvrRpv3.1* and *AvrRpv12* are affected by deletions of different lengths that can comprise many genes in virulent alleles. The gene content fluctuates widely in the *S-AvrRpv10* locus, and large structural rearrangements are observed between parental haplotypes. In the *AvrRpv12* locus, CNV of RXLR genes occurs in both virulent and avirulent alleles, suggesting an evolution driven by repeated tandem duplications, possibly mediated by transposable elements and/or unequal crossovers [29,59]. Notably, this region exhibits a relatively high recombination rate (27.5 vs 18.0 cM/Mb on average in the genome). By contrast, large structural variations, such as those observed at the *S-AvrRpv10* locus, may hinder meiotic recombination when the divergence reaches a critical threshold. This could explain in part why we previously found that putative effector genes were enriched in poorly recombining regions of the *P. viticola* genome [38].

Overall, these observations are reminiscent of the genomic organization of other oomycete pathogens, in which effector genes tend to be located in highly dynamic regions [60–62]. This fast gene turnover in virulence-associated loci fosters a strong adaptative potential, in line with the two-speed genome model [29]. Thus, effectors recognized by the plant can be lost, while new duplicated sequences can diverge to escape recognition or acquire new functions. The genetic diversity in terms of presence-absence variation of effectors is probably one of the keys of the rapid adaptation of *P. viticola* populations to grapevine immune responses [30].

### Parallel adaptation to *Rpv12* occurred in different regions by loss of avirulence

The loss of an Avr factor is one of the most common ways through which oomycete plant pathogens gain virulence [5]. In *P. viticola*, the absence of effector genes is necessary for the breakdown of *Rpv3.1* [36]. Our findings suggest that the mode of adaptation to *Rpv12* is similar and probably linked to the loss of one or several effector(s). Both resistance breakdowns thus appear to be standard cases of ETI evasion. This is consistent with the dominance of avirulence, as a single Avr copy is sufficient to trigger HR and limit pathogen development.

Interestingly, we observed two types of virulence alleles at the *AvrRpv12* locus. Deletions (avr$^{\Delta 1-3}$ alleles in Fig 1D) are associated with the complete absence of HR. By contrast, the avr$^{ns}$ allele produces a high level of sporulation but also visible necrosis in the area of pathogen growth. This "trailing necrosis" phenotype is commonly observed in partial resistance to downy mildew in *Arabidopsis thaliana* and it is generally interpreted as the result of a late or weaker immune response [63,64]. Incomplete resistance to *Bremia lactucae* was also associated with a delayed HR in lettuce [65]. The

analysis of the 412x2543 cross showed that $avr^{ns}$ is recessive over the avirulent allele, but dominant over the fully virulent one ($Avr > avr^{ns} > avr^{\Delta1}$). Although we cannot completely rule out that minor loci may be involved, the aneuploid offspring cPv462_1 confirmed that a strain carrying only $avr^{\Delta1}$ was fully virulent.

Three paralogous RXLR genes are consistently absent in deletion alleles, raising the question of whether the loss of several effectors is needed to escape Rpv12 recognition. The $avr^{ns}$ allele contains a non-sense mutation in a single RXLR gene, P17g40, whose function is probably severely impaired by the premature termination. However, an incomplete immune response may still be triggered upon recognition of P17g38 and/or P17g39. Given that the introgressed *Rpv12* locus contains several NLR genes [22,66], each candidate effector might be recognized by a different R protein. Alternatively, a truncated P17g40 protein may still be sufficient to initiate the Rpv12-mediated HR, albeit with a delayed response due to the poor interaction. In the case of the flax rust effector AvrM, which directly interacts with its cognate resistance protein, recognition is based on the C-terminal domain, but proteins with large truncations outside of this domain induce weaker cell death [67].

The occurrence of distinct virulence alleles in different winegrowing regions suggests that independent mutational events affecting *AvrRpv12* were selected. Moreover, *Rpv12*-breaking strains exhibited signs of a selective sweep at the locus, in line with the recent introduction of *Rpv12*-carrying varieties. At the regional level, the adaptive potential of *P. viticola* is thus sufficient to enable the rapid selection of Avr alleles escaping recognition, as was observed upon the deployment of *Rpv3.1* [68].

## A suppressor activity could play a role in virulence towards *Rpv10*

Unlike *Rpv3.1* and *Rpv12* breakdowns, we found that virulence towards *Rpv10*-carrying plants is determined by a dominant locus. In diploid or dikaryotic pathogens, virulence is typically a recessive trait because a single avirulent allele is sufficient to trigger an immune response. However, seminal works on the flax rust pathosystem have described an Inhibitor (I) locus capable of preventing recognition of several Avr factors [69,70]. In several plant pathogens, certain effectors can indeed suppress the avirulence conferred by other genes. For example, the *Leptosphaeria maculans* effector AvrLm4–7 masks the presence of AvrLm3 which is normally recognized by Rlm3 [71]. A similar relationship exists between SvrPm3 and AvrPm3 in *Blumeria graminis* [72], the first suppressing the Pm3-mediated recognition of the second. In the downy mildew pathogen *Hyaloperonospora arabidopsidis*, the secreted protein S-HAC1 suppresses the avirulence conferred by HAC1 [73]. Thus, one of the putative effectors exclusive to the active *S-AvrRpv10* haplotype may effectively mask a hypothetical AvrRpv10 factor. The extensive repertoire of secreted proteins in *P. viticola* may have evolved, at least in part, to allow certain effectors to "protect" others by disrupting NLR-mediated immune responses [74].

The interaction appears to be specific to the Rpv10-mediated response, as we found no effect of the QTL on plants carrying *Rpv3.1*, *Rpv12* or no resistance genes. Moreover, the donor strain Pv1419_1 is not aggressive towards *Rpv1* either [35]. However, the active *S-AvrRpv10* allele was not sufficient to totally prevent the HR in Pv1419_1 descendants, whereas little to no necrosis was observed with their parent. The wild strain Pv7219_1 also clearly induced necrosis despite being homozygous for this allele. Similarly, Heyman et al. [34] reported a German isolate highly aggressive towards *Rpv10* which triggered abundant necrosis. Several hypotheses may explain this incomplete suppression of the immune response. Undetected QTLs could contribute to the gain of virulence. As the *Rpv10* locus contains several NLR genes [21], S-AvrRpv10 may mask only one protein among several recognized effectors. Alternatively, the suppression activity may vary between strains due to different expression levels, as observed for the *B. graminis* effector SvrPm3 [75].

## The breakdown of *Rpv10* was facilitated by admixture

In this study, we showed that adaptive admixture enabled the emergence of virulence towards *Rpv10* in Europe. This finding aligns with a broader pattern observed in several plant pathogens where introgression between divergent populations, and sometimes different (sub)species, has facilitated adaptation to new hosts and the evolution of distinct pathotypes

 

[9,76,77]. A notable example is the breakdown of apple scab resistance gene *Rvi6* that occured after crabapple-associated *Venturia inaequalis* strains invaded orchards and later hybridized with the agricultural population, leading to the introgression of the virulent trait [8,78].

Invasive pathogens are generally characterized by a reduced genetic diversity in their new habitat, due to the genetic drift resulting from the small initial population size. This founder effect is clearly observed in *P. viticola* populations outside of North America [10]. Consequently, as the center of origin of *P. viticola*, this continent harbors an allelic diversity that is not fully represented in the rest of the world. It is therefore plausible that recently introduced American genotypes may have brought beneficial variants that facilitated the adaptation to *Rpv10*. In accordance with this hypothesis, we detected the candidate effector S10-SP6 in some of the isolates sampled on wild and cultivated *Vitis* across the Eastern United States, where it seems to show presence-absence variation. As *Rpv10* was introgressed from the Asian species *V. amurensis*, these unexpected results suggest that plant pathogens can indeed adapt to plant resistances through the introduction of genetic diversity from their center of origin.

Nevertheless, our limited sample size may bias the genetic clustering analyses. The admixture signal could be incorrectly attributed to the North American cluster because the actual source population may be insufficiently sampled or absent from the panel [79,80].

In any case, the detection of a new entry of the pathogen into Europe associated with resistance breakdown is particularly alarming. This reinforces the view that secondary introductions of already established invasive pathogens should be avoided as this can expand their allele reservoir and thus their adaptive potential [7,81]. In the case of grapevine downy mildew, pathogen movement from North America is especially concerning, because several *Plasmopara* species native to the region can infect wild and cultivated grapes [58,82].

## Conclusion

Overall, we demonstrated that virulence towards partial resistance genes was determined by major loci and we identified promising putative effectors involved in the interaction of *P. viticola* with Rpv10 and Rpv12. By integrating assessments of sporulation and necrosis, we were able to precisely characterize the extent of resistance breakdowns, whether partial or complete. We thus confirmed that gene-for-gene relationships are not restricted to complete resistances and that this model remains relevant in the context of quantitative interactions across various pathosystems [83,84]. We also identified a potential suppression activity targeting the Rpv10-mediated response, which could add another layer of complexity to the molecular interplay. Future functional assays will help decipher the role of candidate effectors, a challenging task for an obligate biotrophic pathogen like *P. viticola*. Co-expression of resistance and effector gene pairs would also require the cloning of Rpv10 and Rpv12, which to our knowledge has yet to be achieved.

Preserving the durability of resistances is critical for perennial plants such as grapevines, which are planted for decades with no possibility of crop or varietal rotation. Here, we show that linkage mapping and population genomics can be combined to understand the diverse pathways by which specialized plant pathogens acquire virulence. In *P. viticola*, parallel adaptation can occur independently in different established populations, while punctual admixture events can also contribute to the emergence of virulence. Ensuring the long-term effectiveness of grapevine resistances will require accounting for the multiple trajectories of pathogen adaptation.

## Supporting information

**S1 Fig. Notation of necrosis score on grapevine leaf discs infected by *P. viticola*.** Necrosis patterns on leaf discs were evaluated at 6 dpi by assessing their size, shape and color. Higher necrosis scores correspond to more efficient immune responses. Photos were taken by the authors. Scores were attributed based on the scale proposed by Paineau et al. [35].
(TIFF)

**S2 Fig. QTL mapping of the two *P. viticola* F1 progenies inoculated on susceptible or resistant grapevine culti-vars.** One map was obtained for each parent of each cross. The detection of a QTL in one of the parental maps indicate a phenotypic difference in the progeny depending on which marker alleles were transmitted by the parent. The black dashed line indicates a LOD significance level of 3.2 which was the lowest threshold value determined across the different QTL mappings ($\alpha = 0.05$). LOD values were computed based on the percentage of sporulation area at 6 dpi.
(TIF)

**S3 Fig. Confirmation of the *AvrRpv3.1* locus in a *P. viticola* biparental population.** (a) Phenotypes distribution in the 412x2543 F1 progeny (N = 162) on cv. 'Regent' (Rpv3.1). (b) QTL mapping of Rpv3.1-breakdown in the linkage map of the avirulent parent Pv2543_1. The gray area indicates the credible interval of the QTL. Dashed lines indicate the LOD significance thresholds determined using 1000 permutations. Results on other linkage groups and other cultivars are available in S2 Fig. (c) Distribution of the sporulation area on Rpv3.1 depending on the inherited allele at the QTL. Horizontal lines in the boxplots signal the 25th, 50th and 75th percentiles. (d) Allelic configurations of the parent strains in the QTL, which corresponds to the same region previously identified by GWAS [36]. The marker corresponding to the peak of the QTL in the present study is indicated by an orange bar on the scale. The allele associated with avirulence corresponds to the non-deleted Pv2543_1 haplotype (named Avr on the left). The secreted proteins P14g164 and P14g165 (colored in red) are totally or partially deleted in the virulent haplotypes.
(TIF)

**S4 Fig. Maximum likelihood phylogenetic tree of RXLR protein sequences around the *AvrRpv12* locus.** Boot-strap support values obtained from 1000 replicates are indicated for each node. RXLR genes absent in all or some Rpv12-breaking strains are highlighted in red boxes. The credible interval of the AvrRpv12 QTL on contig Primary_000017F is indicated in turquoise.
(TIF)

**S5 Fig. Coverage at the *AvrRpv12* locus in avirulent and virulent *P. viticola* strains.** Copy number is indicated on the y-axis and calculated along 5 kb windows. At the bottom, coding sequences are indicated in black, or in red for RXLR genes. The reference avirulent strain Pv221_1 is at the top. Blue boxes signal strains virulent on *Rpv12*. Some strains present higher coverage for the fourth and fifth RXLR genes, suggesting they possess additional copies. Note the hemizy-gous profile of strain Pv2963, which was collected from the same plot as Pv2543_1 but is avirulent.
(TIF)

**S6 Fig. Predicted tertiary structure of an AvrRpv12.** Candidate protein P17g40 code for an RXLR protein composed of several modules of the LWY domain. (a) Complete AlphaFold-predicted structure colored by predicted local distance difference test (pLDDT) (red: low confidence, blue: high confidence). The N terminus and C terminus of the molecule are indicated by 'Nter' and 'Cter'. (b) Predicted aligned error (pAE) of the relative position of residues along the protein sequence. (c) Predicted structures with the 9 complete LWY modules highlighted in different colors. Sequences linking the different modules are shown in blue. Poorly predicted N- and C-terminal parts were trimmed for visual clarity. (d) Super-imposition of all LWY domains of P17g40, using the third domain from the oomycete effector PsPSR2 as a reference. (e) Alignment of the P17g40 and the PsPSR2 sequences, skipping the first module that is shorter than the others in both proteins [26]. Green lines at the top of the alignment indicate alpha-helices sequences for P17g40, and purple lines at the bottom indicate those of PsPSR2. Conserved leucine residues contributing to the fold are highlighted in red. Black back-ground indicate identity and gray background similarity (70% cutoff).
(TIFF)

**S7 Fig. Signs of a recent selective sweep around the *AvrRpv12* locus in virulent *P. viticola* strains.** Blue boxes indi-cate the *AvrRpv12* QTL and grey boxes show the limits of contig Primary_000017F. (a) Runs of Homozygosity (ROH)

on chromosome 16. Lines indicate uninterrupted homozygous segments. Avirulent and virulent strains from three different geographical origins are shown. The VB strain was previously studied in Wingerter et al. [18]. Plot made using r/detectRUNS. (b) Cross-population Extended Haplotype Homozygosity (XP-EHH) calculated by Selscan along chromosome 16. Virulent strains from Hungary (n = 6) were compared to avirulent strains from the same country (n = 9). The highest positive scores are observed around *AvrRpv12*, suggesting positive selection in the virulent subpopulation. (TIF)

**S8 Fig. Cross-validation error estimates from the genetic clustering analyses of the *P. viticola* diversity panel They were computed by the ADMIXTURE program with option –cv = 5.** The value is lowest for K = 3, making it the best-fitting number of clusters. (TIF)

**S1 Methods. Additional information on the chromosome-level assembly of strain Pv1419_1, the genotyping of the backcross population by amplicon length polymorphism, and variant calling and filtration for the population structure analyses.** (DOCX)

**S1 Table. Assembly statistics of the Pv1419_1 genome.** (XLSX)

**S2 Table. Primers and PCR conditions used to determine *S-AvrRpv10* genotypes.** (XLSX)

**S3 Table. Primer and probe sequences for the duplex qPCR assay targeting PvTUB and S10-SP6.** (XLSX)

**S1 Data. *P. viticola* strains referenced in the study.** (XLSX)

**S2 Data. Gene annotation in the AvrRpv12 locus.** (XLSX)

**S3 Data. Genes encoding putative secreted proteins in the S-AvrRpv10 locus.** Expression levels are provided for 24, 48 and 72 hours post-inoculation by the avirulent strain Pv221_1. (XLSX)

**S4 Data. Presence-Absence Variation of genes encoding putative secreted proteins in the two Pv1419_1 haplotypes.** (XLSX)

**S5 Data. Genotypes at the S-AvrRPv10 locus obtained by amplicon length polymorphism.** (XLSX)

**S6 Data. Detection by qPCR of the candidate effector gene S10-SP6 in DNA samples collected in the United States.** (XLSX)

## Acknowledgments

We thank Julie Bourg, Carole Couture and Anne-Sophie Miclot (INRAE Bordeaux, France) for technical assistance, Frédéric Fabre for his advice on statistical analyses and Sabine Wiedemann-Merdinoglu (INRAE Colmar, France) for her insights on necrosis notation. We are grateful to the people who provided us with *P. viticola* strains: Pál Kozma (University

of Pécs, Hungary), Lance Cadle-Davidson and Anna Underhill (Cornell University, Geneva, NY, United States), Odile Carisse (Horticultural Research and Development Centre, Agriculture and Agri-Food Canada), René Fuchs and Stefan Schumacher (State Institute of Viticulture and Oenology, Freiburg im Breisgau, Germany), Jochen Bogs, Birgit Eisenmann and Chantal Wingerter (Winecampus Neustadt, Germany), Elisa De Luca (Vivai Cooperativi Rauscedo, Rauscedo, Italy). We acknowledge the Genotoul bioinformatics platform (Toulouse, France) for providing computing and storage resources.

## Author contributions

**Conceptualization:** Etienne Dvorak, Marie Foulongne-Oriol, François Delmotte.

**Formal analysis:** Etienne Dvorak, Thomas Dumartinet, Pere Mestre, Marie Foulongne-Oriol, François Delmotte.

**Funding acquisition:** François Delmotte.

**Investigation:** Etienne Dvorak, Thomas Dumartinet, Isabelle D. Mazet, Alexandre Chataigner, Manon Paineau, Dario Cantù, Pere Mestre, Marie Foulongne-Oriol, François Delmotte.

**Resources:** Manon Paineau, Dario Cantù.

**Supervision:** Marie Foulongne-Oriol, François Delmotte.

**Visualization:** Etienne Dvorak, Thomas Dumartinet, Pere Mestre.

**Writing – original draft:** Etienne Dvorak, Marie Foulongne-Oriol, François Delmotte.

**Writing – review & editing:** Etienne Dvorak, Thomas Dumartinet, Pere Mestre, Marie Foulongne-Oriol, François Delmotte.

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
