## [Decision Letter · Decision Letter 0]

19 Sep 2025

Parallel adaptation and admixture drive the evolution of virulence in the grapevine downy mildew pathogen

PLOS Pathogens

Dear Dr. Delmotte,

Thank you for submitting your manuscript to PLOS Pathogens. After careful consideration, we feel that it has merit but does not fully meet PLOS Pathogens's publication criteria as it currently stands. Therefore, we invite you to submit a revised version of the manuscript that addresses the points raised during the review process.

Please submit your revised manuscript within 60 days Nov 18 2025 11:59PM. If you will need more time than this to complete your revisions, please reply to this message or contact the journal office at plospathogens@plos.org. Please include the following items when submitting your revised manuscript:

We look forward to receiving your revised manuscript.

Kind regards,

Eva H. Stukenbrock, PhD

Academic Editor

PLOS Pathogens

Savithramma Dinesh-Kumar

Section Editor

PLOS Pathogens

Editor-in-Chief

PLOS Pathogens

orcid.org/0000-0003-2946-9497

Editor-in-Chief

PLOS Pathogens

orcid.org/0000-0002-7699-2064

**Additional Editor Comments:**

Below the three reviewers have provided constructive comments and suggestions to your manuscript. Among several comments, the reviewers were not convinced about the conclusions from your admixture analysis. Please make sure to consider the limitations of your study/sampling and relate your findings to previous literature. Reviewer 1 also commented on your inference of selection based on patterns of homozygosity. As mentioned, additional methods are necessary to strengthen the conclusion related to selection acting on genes in the QTL.

I kindly invite you to prepare a revised version of the manuscript that addresses every comment point-by-point.

**Journal Requirements:**

https://journals.plos.org/plospathogens/s/submission-guidelines#loc-parts-of-a-submission

- ® on page: 23.

5) We have noticed that you have uploaded Supporting Information files, but you have not included a list of legends. Please add a full list of legends for your Supporting Information files after the references list.

Potential Copyright Issues:

- Please confirm (a) that you are the photographer of Figure S1., or (b) provide written permission from the photographer to publish the photo(s) under our CC BY 4.0 license.

7) Please ensure that the funders and grant numbers match between the Financial Disclosure field and the Funding Information tab in your submission form. Note that the funders must be provided in the same order in both places as well.

**Reviewers' Comments:**

Reviewer's Responses to Questions

**Part I - Summary**

Reviewer #1: P. viticola is a devastating disease of grapevines, which necessitates extensive fungicide applications. Therefore, investigating resistance and the breakdown of resistance in grapevines is a crucial topic in this field. In this paper, the authors employ a genetic approach using two different mapping populations of P. viticola to explore the genetic basis of virulence on grapevines varieties carrying the Rpv10 and Rpv12 resistance genes. Using QTL mapping, they identified a single locus underlying virulence on both gene. For Rpv12, the identified virulence allele displayed recessive behaviour, consistent with a dominant Avr factor in this locus. In the case of Rpv10, the authors provide evidence that it functions as a dominant suppressor locus, rather than containing the actual avirulence gene.

There is evidence of structural rearrangements at both loci, including presence/absence polymorphisms of bona fide effector genes from downy mildew. Following the locus definition, the authors also aim to investigate the evolutionary scenario under which resistance breakdown occurred, and provide evidence that, at the Rpv12 locus, resistance breakdown happened in several locations, followed by a selective sweep. In the case of Rpv10, adaptive introgression from a strain from the USA seems to be the underlying mechanism.

Due to the continuous threat of grapevine downy mildew to wine production, and the ecological consequences of fungicide resistance as a result, this study, which provides insight into the resistance breakdown of two resistance loci in grapevine, represents an important contribution. The study by Dvorak et al. is a beautiful example of how pathogen genetics can be used to dissect complex AVR loci and understand resistance breakdown. At certain points in the manuscript, a bit more technical detail on how the bioinformatic analyses were carried out would be beneficial for reproducibility. The analysis concerning the evolutionary scenario would also benefit from additional analysis or considertions. Overall, the manuscript is well written, clearly structured, and I really enjoyed reading it.

Reviewer #2: This is an excellent investigation of the evolution of virulence in the grapevine downy mildew pathogen. The manuscript is well-written and detailed. The analyses are comprehensive, and the results are fascinating and of great concern regarding durability of resistance.

The final analysis of the distribution of variants in sequenced samples from different regions was the only analysis that felt incomplete, because it appears that the North American isolates were not phenotyped on Rpv10 plants.

Reviewer #3: Dvorak et al. use robustly designed genetic experiments to map avirulence/virulence on different grapevine downy mildew resistance genes in P. viticola bi-parental populations. Using a pseudo-testcross mapping strategy, the authors confirmed previous mapping results on the AvrRpv3 locus, as well as confidently mapped loci involved with avirulence/virulence on the Rpv10 and Rpv12 resistance genes. A very high level of genomic plasticity was observed underlying the mapped loci, including substantial structural variations and gene duplications involving genes encoding predicted secreted proteins with RXLR motifs. Of note, the locus implicated in partially breaking the Rpv10 resistance gene appears to function as a suppressor. Finally, a population genomic approach was taken to evaluate local ancestry across the genome and specifically, at the S-AvrRpv10 locus, revealing that partial virulence on Rpv10 was likely introduced into European strains by crossing with an individual closely related to strains collected in North America. Overall, the authors combined elegant genetic experiments, advanced genomics methods, and population biology concepts to shed light on the complex dynamics of avirulence/virulence in the P. viticola – grapevine pathosystem. The results are novel and will be of general interest to the journal’s audience. I have a few minor comments for the authors’ consideration below.

**Part II – Major Issues: Key Experiments Required for Acceptance**

Please use this section to detail the key new experiments or modifications of existing experiments that should be absolutely required to validate study conclusions.required to validate study conclusions.

Reviewer #1: 1) The authors conclude that long stretches of homozygosity are indicative of recent selective sweeps. While I agree that this is a plausible indication, I believe additional analyses would strengthen this conclusion. For example, do strains that do not show virulence breakdown display a different pattern of homozygosity? One would expect this to be the case if the selective sweep was acting specifically on the virulent allele. Additionally, the authors should consider more formal approaches to detect selective sweeps. For instance, using tools like isoRelate to identify identity-by-descent (IBD) segments could provide more robust evidence for recent sweeps.

2. On a related note, the claim based on the admixture analysis could benefit from additional consideration and perhaps some discussion of the existing literature. For example, Fountain et al., 2021, which is cited throughout the discussion, did not detect any admixture between the American and European populations, but rather between the European and South African populations. In, other words, could the relatively limited number of non-European isolates in the current study be skewing the admixture analysis? A discussion of this limitation, and how it might affect the interpretation of the results, should be included.

Reviewer #2: Were all the strains in Figure 4A phenotyped on Rpv10 or only the ones identified as resistance breaking? Is the introgressed haplotype in the isolates that have overcome Rpv10 identical to the sequenced isolates from North America or is the ADMIXTURE analysis clustering them because the actual source population was not sampled? For example, Fontaine et al. 2021 genotyped isolates from China and the authors mention that the Rpv10 resistance was sourced from East Asia. Could it be that the introgression was from an unsampled V. aestivalis population? I don’t question the introgression but I wonder about the inference that the US is the source given the limited sampling of other populations and their close relationships. The authors sometimes qualify the inference, for example “extra-European population” on line 414, but the discussion is more definitive about a secondary introduction from North America.

Reviewer #3: (No Response)

**Part III – Minor Issues: Editorial and Data Presentation Modifications**

Reviewer #1: It personally took me a while to understand why there are genetic maps for both parental isolates separately. This is probably very obvious to someone working with a diploid species, but for those of us who don’t, a little explanation would certainly help with understanding the manuscript.

Also, at the beginning of the Results section (around Line 225), it might be good to restate what kind of population these are.

Line-specific comments:

Line 136: Please include a reference to the supplementary figure showing the scale.

Line 155: A bit more detail would be helpful. What was the command used to run the analysis—was it scanone? Same question for composite interval mapping.

Line 216: What subset of the total number of SNPs was considered “fixed” in this analysis?

Line 217: Which strains were defined as pure “European” ancestry, given that Admixture identifies two ancestries for Europe at K=3, correct?

Line 245: High levels of sporulation tended to be associated with light necrosis surrounding sporulation spots (average necrosis score < 3). Where is this data shown?

Line 256: Which reference genome was used?

Line 257: What is the gene name?

Line 296: What is the sporulation phenotype of this isolate?

Line 382: What is the CV error? Where is this data shown?

Data availability:

Maybe you could extend the data availability statement, saying that not only genome and annotation are available through the provided link, but also phenotypes and map data.

Supplementary Methods and Figures:

Methods S1: This section is lacking a bit of technical depth. How was the hifiasm program run (with what specifications)? What was the threshold for keeping a contig? With what parameters was hifiasm run? How was the genetic map used to assemble the contigs? Was short-read data used to phase the genomes?

Methods S2: Please indicate where the primer sequences can be found.

Methods S3: How were repetitive regions defined? Did you perform repeat masking with a repeat library?

Figure S6: What does the yellow box indicate?

Reviewer #2: 1. I advise replacing “pure” in reference to European and North American samples (line 403). Given that the European samples are descendants of the North American population, I’m not sure this is an appropriate term. And it is problematic in the larger context of population genetics, so best to not use it and instead refer to SNPs associated with ADMIXTURE clusters.

2. That the RXLR motif can take alternative forms, particularly in downy mildews: RXLK, RVRN, QXLR, GKLR, etc. For the proteins that have the dEER motif, do these have an RXLR-like motif? This information would be useful for computational identification of cytoplasmic effectors in other Plasmopara, especially if those that don’t have dEER or LWY.

3. The three North American isolates sequenced were identified as belonging to “clade aestivalis, the only species present worldwide” (line 191). The reader must infer, or look at Fontaine et al. 2021, to determine that the strains being examined from Europe are also clade aestivalis.

4. Tandem is used as a noun but I typically think of it as an adjective – tandem array of genes or tandem duplicate genes.

5. I suggest point mutations or nucleotide substitutions instead of “punctual variants”.

Reviewer #3: L104-107: Although it is mentioned later in the manuscript, stating the genotypes of the cPv44_ 1 and Pv1419_1 strains here when mentioning the backcross population would help make the objective/rationale for this experiment clearer.

-In Methods S1, it is stated “The ten most promising assemblies were then screened for contamination.”. What criteria were used to select these ten assemblies?

Fig. S4 – The methods used in this analysis are not presented in the manuscript.

Fig. S5 – Methods also missing for the read depth analysis.

L394-396: Could the lower level of admixture observed in the Rpv10-breaking strains besides Pv1419_1 be caused by recurrent backcrossing to the Upper Rhine Plain population? There also seems to be decent separation between these three Rpv10-breaking strains and the rest of the cluster in the lower-left of the PCA plot.

PLOS authors have the option to publish the peer review history of their article (what does this mean? ). If published, this will include your full peer review and any attached files.). If published, this will include your full peer review and any attached files.

**Do you want your identity to be public for this peer review?** For information about this choice, including consent withdrawal, please see our For information about this choice, including consent withdrawal, please see our Privacy Policy ..

Reviewer #1: No

Reviewer #2: No

Reviewer #3: No

**Figure resubmission:**

**Reproducibility:**



---

## [Decision Letter · Decision Letter 1]

11 Feb 2026

PPATHOGENS-D-25-01551R1

Parallel adaptation and admixture drive the evolution of virulence in the grapevine downy mildew pathogen

PLOS Pathogens

Dear Dr. Delmotte,

Thank you for submitting your manuscript to PLOS Pathogens. After careful consideration, we feel it has merit, but we ask for a few edits to ensure it is ready for acceptance by PLOS Pathogens. Therefore, we invite you to submit a revised manuscript that addresses the points raised during the review process.

We look forward to receiving your revised manuscript.

Kind regards,

Eva H. Stukenbrock, PhD

Academic Editor

PLOS Pathogens

Savithramma Dinesh-Kumar

Section Editor

PLOS Pathogens

Sumita Bhaduri-McIntosh

Editor-in-Chief

PLOS Pathogens

orcid.org/0000-0003-2946-9497

Michael Malim

Editor-in-Chief

PLOS Pathogens

orcid.org/0000-0002-7699-2064

**Additional Editor Comments:**

The reviewers all agree that all concerns were raised satisfactorily. I have just a few minor points before the manuscript can be finally accepted. These small text and formatting-related issues should be addressed.

The abstract should just be one paragraph, so please remove line breaks.

Please revise and improve text in Author Summary including:

Line 37 insert “…the resistance genes…”

Line 39 Enriched in stead of rich

Line 40 something is missing “…One??”

Intro:

Line 92: Please change: “Thanks to recent advances in the sequencing and annotation of the genome of P. viticola, the first Avr locus was identified.….”

Methods:’

Line 250: “….in 5 kb windowns….”

Results:

Line 289: is “credible interval” correct here? I think the term mainly used in Bayesian analyses. Shouldn’t it be “confidence interval” here? And in Figure legend 1 and 2 and supplementary figures? Please check which term is correct according to the analysis used.

Line 460: Please insert: “… recurrent backcrossing with the local P. viticola population”

Methods S1-S3: I recommend joining these files into one supplementary file “Supplementary Methods”

Please insert title of manuscript and author names on first page of the document.

**Journal Requirements:**

**Reviewers' Comments:**

Reviewer's Responses to Questions

**Part I - Summary**

Reviewer #1: The authors have thoroughly addressed all of my previous comments. I particularly appreciate their effort to provide additional evidence supporting a selective sweep at AvrRpv12 and to include a more careful discussion of the ancestry results. In addition, the revised manuscript shows substantial improvements in methodological details.

Reviewer #2: The authors did a nice job addressing the reviewer comments on the first submission, including collecting additional data.

Reviewer #3: The authors have sufficiently addressed all previous comments and concerns. I have no further suggestions or comments.

**Part II – Major Issues: Key Experiments Required for Acceptance**

Please use this section to detail the key new experiments or modifications of existing experiments that should be absolutely required to validate study conclusions.required to validate study conclusions.

Reviewer #1: (No Response)

Reviewer #2: None

Reviewer #3: (No Response)

**Part III – Minor Issues: Editorial and Data Presentation Modifications**

Reviewer #1: (No Response)

Reviewer #2: None

Reviewer #3: (No Response)

PLOS authors have the option to publish the peer review history of their article (what does this mean? ). If published, this will include your full peer review and any attached files.). If published, this will include your full peer review and any attached files.

**Do you want your identity to be public for this peer review?** For information about this choice, including consent withdrawal, please see our For information about this choice, including consent withdrawal, please see our Privacy Policy ..

Reviewer #1: No

Reviewer #2: No

Reviewer #3: No

**Figure resubmission:**
---

## [Editor Report · Decision Letter 2]

26 Feb 2026

Dear Dr. Delmotte,

We are pleased to inform you that your manuscript 'Parallel adaptation and admixture drive the evolution of virulence in the grapevine downy mildew pathogen' has been provisionally accepted for publication in PLOS Pathogens.

Best regards,

Eva H. Stukenbrock, PhD

Academic Editor

PLOS Pathogens

Savithramma Dinesh-Kumar

Section Editor

PLOS Pathogens

Sumita Bhaduri-McIntosh

Editor-in-Chief

PLOS Pathogens

orcid.org/0000-0003-2946-9497

Michael Malim

Editor-in-Chief

PLOS Pathogens

orcid.org/0000-0002-7699-2064
---

## [Editor Report · Acceptance letter]

Dear Dr. Delmotte,

We are delighted to inform you that your manuscript, "Parallel adaptation and admixture drive the evolution of virulence in the grapevine downy mildew pathogen," has been formally accepted for publication in PLOS Pathogens.

Best regards,

Sumita Bhaduri-McIntosh

Editor-in-Chief

PLOS Pathogens

orcid.org/0000-0003-2946-9497

Michael Malim

Editor-in-Chief

PLOS Pathogens

orcid.org/0000-0002-7699-2064